# Metal Contact Induced Unconventional Field Effect in Metallic Carbon Nanotubes

**DOI:** 10.3390/nano13111774

**Published:** 2023-05-31

**Authors:** Georgy Fedorov, Roohollah Hafizi, Vyacheslav Semenenko, Vasili Perebeinos

**Affiliations:** 1Institute of Photonics, University of Eastern Finland, 999018 Joensuu, Finland; 2Department of Physics and Astronomy and Thomas Young Centre, University College London, London WC1E 6BT, UK; 3Department of Electrical Engineering, University at Buffalo, The State University of New York, Buffalo, NY 14260, USA

**Keywords:** carbon nanotubes, ballistic transport, electrical contact resistance

## Abstract

One-dimensional carbon nanotubes (CNTs) are promising for future nanoelectronics and optoelectronics, and an understanding of electrical contacts is essential for developing these technologies. Although significant efforts have been made in this direction, the quantitative behavior of electrical contacts remains poorly understood. Here, we investigate the effect of metal deformations on the gate voltage dependence of the conductance of metallic armchair and zigzag CNT field effect transistors (FETs). We employ density functional theory calculations of deformed CNTs under metal contacts to demonstrate that the current-voltage characteristics of the FET devices are qualitatively different from those expected for metallic CNT. We predict that, in the case of armchair CNT, the gate-voltage dependence of the conductance shows an ON/OFF ratio of about a factor of two, nearly independent of temperature. We attribute the simulated behavior to modification of the band structure under the metals caused by deformation. Our comprehensive model predicts a distinct feature of conductance modulation in armchair CNTFETs induced by the deformation of the CNT band structure. At the same time, the deformation in zigzag metallic CNTs leads to a band crossing but not to a bandgap opening.

## 1. Introduction

Single-walled carbon nanotubes (CNTs) have emerged as highly promising materials for next-generation nanoelectronics and optoelectronics [1,2]. Semiconducting CNT-based field effect transistors can achieve efficient gate electrostatics and superior transport characteristics [3,4,5,6,7,8,9]. Metallic CNTs demonstrate high current density capacity for electrical interconnects [10], while small-gap quasi-metallic CNTs can be used for THz optoelectronics [11]. In ultimately scaled short-channel CNT devices, contacts play a crucial role in determining the CNT device’s performance [12,13]. Moreover, CNTs are fascinating because of their unique properties, such as the lack of backscattering in pristine CNTs with a perfect atomic structure. Recent progress in fabrication of novel CNT-based devices [14,15,16] stimulates further research in this direction. The achievement of highly transparent electrical contacts in CNTs is a crucial challenge for implementation in nanoelectronics, and many studies have described new strategies to reduce contact resistance [17]. Although multiple studies have investigated the electrical contacts of CNTs [18,19], the inability to probe the electrical properties of CNTs under metals impedes the quantitative understanding of electrical contacts.

Recent theoretical work has suggested that metal deposition causes substantial deformation of CNTs by collapsing CNTs with opposite sides touching each other [20,21], as in the graphene bilayer. The electronic structure in thus-deformed CNTs was predicted to be a significant modification of the conventional picture resulting from the zone-folding approximation [22], which predicts a finite gap of 0.1–1.0 eV in semiconducting CNTs with mod(n−m,3)≠0, quasi-metallic CNTs with a small gap of a few tens of meV with mod(n−m,3)=0, and truly metallic CNTs with 0 gap with chiral indices n=m [22]. The latter class of CNTs has a one-dimensional Dirac metal band structure with a linear dispersion relation around the charge-neutrality point (CNP). Electrostatic doping is well-known to change the conductance of the semiconducting CNT device by several orders of magnitude even at room temperature [23], while for quasi-metallic CNTs, the effect is not strong [24]. Moreover, charges in the substrate lead to Fermi-level inhomogeneity such that the effective electrical transport gap is reduced in CNTs on subtrates [25]. The minimum conductance as a function of the gate voltage occurs when the CNP and Fermi levels of the nanotube coincide. In the case of truly metallic CNTs, no dependence of conductance on the gate voltage should be observed around the CNP.

In this paper, we employ Density Functional Theory (DFT) calculations of deformed CNTs under metal contacts to demonstrate that the gate voltage dependence of the conductance of truly metallic armchair CNTs is affected by the deformations. We find a temperature-independent field effect in field effect transistor (FET) devices made with a truly metallic CNT (the device configuration is schematically shown in Figure 1a). The ratio of maximal to minimal conductance (ON/OFF ratio) is predicted to be about a factor of two at all low and high temperatures in the armchair CNTFETs. We attribute the simulated behavior to modification of the nanotube band structure under the metals, which is caused by deformation. At the same time, in the zigzag CNT, deformations lead to band crossing but not opening of the bandgap. As a result, the gate voltage dependence of the conductance is much weaker than expected for metallic CNTs.

## 2. Materials and Methods

Our current work is focused on electrical transport in metallic carbon nanotubes in the field effect transistor geometry, as shown in Figure 1a. In the simulations, we assumed a typical thickness of the oxide layer $t_{ox}$ of 300 nm. Large-diameter CNTs with d≈2.5 nm are vulnerable to deformation [20,21]. Therefore, we examined metallic CNTs with different chiralities: armchair (18, 18) with d=2.44 nm and zigzag (30, 0) with d=2.35 nm.

Armchair and zigzag are two different types of carbon nanotube structures, distinguished by their arrangement of carbon atoms along the tube axis. The key difference lies in the orientation of the hexagonal lattice of carbon atoms. Armchair CNTs have their edges formed by parallel lines resembling the armrests of a chair. The chiral vector (n,n) describes armchair nanotubes. Zigzag CNTs have edges formed by alternating lines that resemble a sequence of ‘z’ shapes. The chiral vector (n,0) describes zigzag nanotubes, where *n* represents the number of carbon atom unit cells along the circumferential direction. The difference in the arrangement of carbon atoms leads to different electronic properties and symmetries of the wavefunctions in armchair and zigzag carbon nanotubes. For example, zigzag nanotubes can exhibit metallic or semiconducting behavior depending on whether *n* is divisible by three or not, while armchair nanotubes are generally metallic. The differences in their electronic properties make them suitable for various applications in nanoelectronics and optoelectronics. It is worth noting that carbon nanotubes can also have chiralities other than armchair and zigzag, known as chiral nanotubes. Chiral nanotubes have chiral vectors of the form (n,m), where *n* and *m* are integers.

The curvature-induced band gap is zero in the case of the so-called armchair nanotubes with chirality indices (n,n), where *n* is an integer. We used this effect to investigate how the appearance of a band gap will affect the transport properties of FET devices. The minimal conductance corresponds to the applied gate voltage, which places the Fermi level at the CNP in the nanotube. Note that the Coulomb interactions in our large-diameter CNTs are suppressed [26] and not taken into account in our simulations.

Following Ref. [27], we relax the atomic position of the armchair (18, 18) and zigzag (30, 0) CNTs under the metal. We employ a semi-atomistic model in which the positions of carbon atoms are relaxed in the presence of the metal and substrate, where both are treated by a continuum model, following Ref. [20]. The interatomic interactions of CNT are described by the valence force model [28], in which the stiffness against the misalignment of neighboring orbitals π is adjusted (using parameters $\beta_{p}=0.003566$ eV and $r_{0}=1.43$ Å) to reproduce the bending stiffness of D=1.4eV [20,29,30,31]. The substrate is planar and rigid, and the metal is isotropic. We choose the surface energy $\gamma =12.5\;\mathrm{eV}/{\mathrm{nm}}^{2}$ as for the Pd metal [32,33]. Pd was chosen as the material that ensures the best electrical contact with a CNT, as noted in many experimental studies. The choice of metal defines the work function difference between the CNT midgap energy and the metal, which enters in our model for conductance gate–voltage dependence simulations. A different choice of metal work function would modify the threshold voltage corresponding to the minimum conductance. In addition, the metal-wetting properties of the substrate define the geometry of the deformed CNT. In that respect, a choice of substrate can also play a role, since stronger adhesion of the metal to the substrate would lead to larger deformations and hence stronger changes in electronic structure. We choose the most commonly used SiO${}_{2}$ substrate in our simulations.

## 3. Results and Discussion

We employ the DFT method to calculate the electronic structure of deformed CNTs. DFT is a computational method used to study the electronic structure and properties of atoms, molecules, and solids. It is based on the concept of electron density, where the total energy of a system is expressed as a functional of the electron density rather than solving the many-body Schrödinger equation directly. By solving the Kohn–Sham equations that are derived from DFT, one can obtain the single-particle electronic energy states or band structure. While DFT is computationally efficient, it has certain limitations. DFT relies on approximations for the exchange-correlation functional, which describes the interaction between electrons. The most commonly used approximations, such as the generalized gradient approximation used here, are not exact and can introduce errors. In particular, DFT suffers from certain limitations when it comes to accurately predicting bandgaps, especially in semiconductors and insulators. The bandgap is often underestimated because the standard DFT functionals, such as the generalized gradient approximation, fail to capture the electron self-energy effects that are important for accurate bandgap predictions.

The GW approximation is a post-DFT method that goes beyond the approximations used in DFT to calculate the electron self-energy. It is based on Green’s function theory and considers the electron–electron interactions more accurately. The GW method improves the accuracy of bandgap predictions by incorporating the effects of electron–electron interactions, which are crucial for describing the excitation energies and quasiparticle properties. However, the GW approximation is computationally more demanding than standard DFT. It involves solving the Dyson equation and evaluating the self-energy corrections, which can be time-consuming and require significant computational resources, especially for large unit cells with 72 and 120 carbon atoms used in our work. Nevertheless, since our DFT calculations predict the bandgap opening in the deformed armchair CNT under the metal, the more accurate GW approximation would predict even larger bandgap values and, as a result, even stronger modulation of the current-gate voltage dependence. One should also keep in mind that metal introduces a screening of the electron–electron interactions, which implies that the DFT result may be closer to the experimental value than the GW approximation. Unfortunately, direct measurement of the bandgap in CNT under the metals is not possible, and only indirect evidence from the current-gate voltage characteristics predicted here can shine a light on the electronic structure of CNTs under the metals.

Our simulation results used two metallic CNT conduction channels, and we predict that in the case of armchair CNT, the gate voltage dependence of the conductance shows an ON/OFF ratio of about a factor of two, nearly independent of temperature. To understand such behavior, we argue that modifications in the nanotube band structure under the metal are responsible for this unusual phenomenon. At the same time, we find that metal deformations in zigzag CNT do not lead to a strong modulation of the conductance with the gate voltage because of the different symmetries of the wavefunctions near the Fermi level.

The relaxed geometry of (18, 18) CNT is shown in Figure 1b, which is used as input for electronic structure calculations implemented in the WIEN2k code [34]. WIEN2k iteratively solves the Kohn–Sham equations self-consistently for the electron density. It employs the augmented plane wave method to expand the wavefunctions in plane waves and atomic-like basis functions to accurately represent the electronic wavefunctions and electron density for all electrons, including core states. This requires higher computational power as compared to pseudo-potential DFT codes, but it achieves higher accuracy. We used the generalized Perdew–Burke–Ernzerhof gradient approximation for the exchange-correlation potential. The vacuum used to separate CNTs is 7 Å, and the *k* mesh is 1×1×32. Deformation breaks the axial symmetry and lifts the two-fold degeneracy, as shown in Figure 2. Furthermore, we find a small opening of the bandgap of 120 meV at the *K*-point in the armchair CNT in Figure 2a, consistent with previous work [35]. The additional conduction and valence bands appear away from the *K*-point, which doubles the number of conduction modes near the charge neutrality point. At the same time, in zigzag CNT in Figure 2b, we find a band crossing but not a bandgap opening.

The transport properties are determined by the position of the Fermi level in the CNT under the metal, which is controlled by an applied gate voltage. To simulate the transfer gate voltage characteristics for the geometry of the CNTFET shown in Figure 1, one would normally solve the self-solving transport equation and the Poisson equations [36] to obtain a self-consistent charge along the CNT. The oxide thickness and the biasing conditions determine a typical length scale for charge variation. In our approach, we consider a quasi-ballistic regime and calculate the maximum current in the CNT channel by calculating the number of conduction modes *M* at the Fermi level in the CNT under the metal and in the undeformed CNT far away from the metal. In the latter case, the number of modes is independent of energy and is equal to M=4, resulting in the well-known ballistic conductance of CNT $G=4e^{2}/h=155$μS, where *e* is the electron charge and *h* in the Planck constant. Under the metal, doping is determined by a difference in the work function of CNT $W_{CNT}$ and metal $W_{M}$, which we assume for the contact of Pd to be $\Delta W=W_{M}-W_{CNT}=0.35$ eV [37]. When the Fermi level falls into the bandgap, the number of modes equals zero, and the conductance vanishes because there are no states at the Fermi level under the metal that contribute to transport. As a result, we predict modulation of the conductance as a function of the gate voltage.

To describe charge density on CNT under the metal as a function of gate voltage, we consider capacitive coupling to the metal electrode $C_{M}$ and to the gate $C_{g}$ (For the metal-CNT capacitance $C_{M}$, we use a planar capacitance model $C_{M}=\epsilon_{0}W/d_{M}$ with the effective CNT width W=4.0 nm, corresponding to the geometry of the deformed CNT shown in Figure 1a. We account for an electrostatic distance $d_{M}=2.5$ Å lower than the van der Waals distance between the metal and the vacuum space of the CNT to reflect the spatial extent of the carbon orbitals $p_{z}$ and *d* [27]. For gate capacitance $C_{g}$, we use a graphene nanoribbon capacitance [38], that is, $C_{g}=\epsilon_{ox}\epsilon_{0}\frac{W}{t_{ox}}k$, where the enhancement factor $k={\left(\frac{2}{\pi }\arctan\frac{W}{4t_{ox}}+\frac{W}{4\pi t_{ox}}\ln\left(1+\frac{16t_{ox}^{2}}{W^{2}}\right)\right)}^{-1}$). We solve self-consistent equations for the charge density in CNT under metal as a function of the gate voltage $V_{g}$:(1)$$\begin{array}{ccc} \rho_{M} & = & -C_{g}\left(V_{g}-V_{CNT}\right)+C_{M}V_{CNT}\\ & = & \int_{-\infty }^{E_{CNP}}dED_{h}\left(E\right)\left(1-f_{0}\left(E,E_{F}\right)\right)\\ & - & \int_{E_{CNP}}^{\infty }dED_{e}\left(E\right)f_{0}\left(E,E_{F}\right), \end{array}$$
where $f_{0}\left(E,E_{F}\right)$ is the Fermi–Dirac function, and $E_{F}$ is the Fermi energy. The electron and hole density of the states $D_{e}$ and $D_{h}$ are found using the Density Functional Theory computational modeling method from the band structure shown in Figure 2 according to $D\left(E\right)=1/\left(N_{k}L\right)\sum_{k,i}\delta \left(E-E_{i}\left(k\right)\right)$, where the sum runs over discretized $N_{k}$ k-points in the Brillouin zone and band index *i*, *E* is the energy, δ(x) is a delta function of argument *x*, which we implement numerically using a Gaussian function of finite width, E(k) is the electronic band structure shown in Figure 2, and *L* is the length of CNT. In the case of linear dispersion in an undeformed CNT $E\left(k\right)=\hbar v_{F}k$, where $v_{F}\approx 0.86\times 10^{8}$ cm/s is the Fermi velocity in metallic CNT in DFT calculations, the density of states is a constant $D\left(E\right)=4/\left(\pi \hbar v_{F}\right)$. Note that in the experiments, $v_{F}\approx 10^{8}$ cm/s. In the semiconducting CNT with bandgap $E_{g}=2\Delta$ and band dispersion $E\left(k\right)=\sqrt{\Delta^{2}+\hbar^{2}v_{F}^{2}k^{2}}$, the DOS shows Van Hove singularity $D\left(E\right)=4E/\left(\pi \hbar v_{F}\sqrt{E^{2}-\Delta^{2}}\right)$.

In the case of metallic CNTs, a typical bandstructure, not shown in Figure 2 to keep it less busy, consists of a doubly degenerate linear band $E\left(k\right)=\pm \hbar v_{F}k$ and a higher-energy four-fold degenerate semiconducting band $E\left(k\right)=\sqrt{\Delta^{2}+\hbar^{2}v_{F}^{2}k^{2}}$ with a value of $\Delta =2\hbar v_{F}/d\approx 0.45$ eV for a CNT with a diameter of d=2.5 nm [1]. Note that, in the deformed CNT, degeneracies are lifted, and the bands shown in Figure 2 are single degenerate. The CNT potential $V_{CNT}$ is found from the alignment of the band energy: $W_{M}=W_{CNT}+eV_{CNT}+E_{CNP}-E_{F}$, where $W_{M}$ and $W_{CNT}$ are the metal and CNT work functions, respectively, and $E_{CNP}$ is the charge neutrality point. Doping determines the number of conduction modes $M=\min(M_{M},M_{ch})$ at the Fermi energy, where $M_{M}$ is the number of modes in CNT under the metal, and $M_{ch}$ is the number of modes in the channel. Since the axial symmetry is preserved in the CNT channel, we use $M_{ch}=2$, which accounts for the double degeneracy of the bands. The number of modes under the metal follows from the DFT calculations (We evaluate a number of conduction modes under the metal using DFT bandstructure calculations $E_{k}$, such that $M\left(E\right)=\int dk\delta \left(E-E_{k}\right)\frac{dE_{k}}{dk}H\left(\frac{dE_{k}}{dk}\right)$, where H(x) is the Heaviside step function, and δ-function is replaced by a Gaussian broadening $G\left(x\right)=\exp\left(-x^{2}/\delta^{2}\right)/(\sqrt{\pi }\delta$)). The gate voltage dependence of the FET conductance can be simulated using the following:(2)$$\begin{array}{ccc} & & G\left(V_{g}\right)=\frac{2e^{2}}{h}\int_{\infty }^{\infty }dET\left(E\right)M\left(E\right)\\ & & \times \int dE_{F}^{\prime }\frac{df_{0}\left(E,E_{F}^{\prime }\right)}{dE}G\left(E_{F}-E_{F}^{\prime }\right), \end{array}$$
where spin degeneracy is included, and the transmission coefficient T(E) accounts for the double barrier from the metal to the CNT and from the CNT under the metal to the CNT in the channel. The Fermi–Dirac function is evaluated under the metal, and we account for the inhomogeneity of the CNP by the Gaussian broadening function *G* using the broadening parameter of δ=70 meV [39]. In Equation (Equation 2), we account for the fact that, due to inhomogeneity, states with different Fermi energies $E_{F}^{\prime }$ centered around $E_{F}$ contribute to electrical transport. The central Fermi energy $E_{F}$ is found from the self-consistent electrostatics from Equation (Equation 1).

Figure 3 shows the results of the simulations for temperatures 300 K and 80 K. We find that in the case of an armchair (18, 18) CNT, the strong dependence of the conductance on the gate voltage should be observed due to the deformation of the CNT under the metal electrodes. Given that the density of states is constant and energy-independent close to the Dirac point, one expects that the conductance of a device considered in this work should not depend on the gate voltage. Therefore, a simple measurement of a transistor curve of an armchair CNT can probe the deformation of the CNT under contacts, as shown in Figure 3a. The asymmetry in the $G(V_{g})$ curve is due to the difference in the work function between the metal and the CNT, such that a finite threshold voltage must be applied to move the Fermi level under the metal into the band gap to reach the minimum conductance. However, in (30, 0) CNT, the conductance is nearly independent of the gate voltage, as shown in Figure 3b. We use a typical value of the transmission coefficient T(E)=0.3, such that the maximum conductance in Figure 3 corresponds to $G=T4e^{2}/h=46.5$μS. Note that a self-consistent solution for electrostatic potential along the CNT channel would introduce energy dependence for the transmission coefficient corresponding to CNT under the metal and CNT in the channel barrier. However, for unipolar transport n−n and p−p, the transmission coefficient does not strongly depend on energy [39], and the main effect of the modulation predicted here is due to the opening of the bandgap under the metal.

Note that in the absence of Gaussian broadening in Equation (Equation 2), the minimum conductance would show the Arrhenius temperature dependence in the armchair CNTFETs. In addition, simulations of low-temperature gate voltage oscillations require accounting for interference effects [40,41,42,43,44,45], which is beyond the scope of this work.

Our findings suggest that the deformation of CNTs can be engineered in the CNT channel by depositing a dielectric on top of a CNT in the channel. This can be achieved by using dielectrics with different wetting properties and surface energies. As a result, novel small-gap CNT-based devices can be realized for optoelectronic applications in the THz frequency range.

## 4. Conclusions

In conclusion, we have shown theoretically that the gate voltage dependence of the conductance in metallic armchair CNTs depends on the modification of the band structure as a result of the metal deformation. We attribute the modulation of the conductance by the gate voltage to the bandgap opening in the CNT portion under the metal. At the same time, we find a band crossing but not a bandgap opening in zigzag CNT because of the different symmetries of the wavefunctions near the Fermi level.

## Figures and Tables

**Figure 1 nanomaterials-13-01774-f001:**
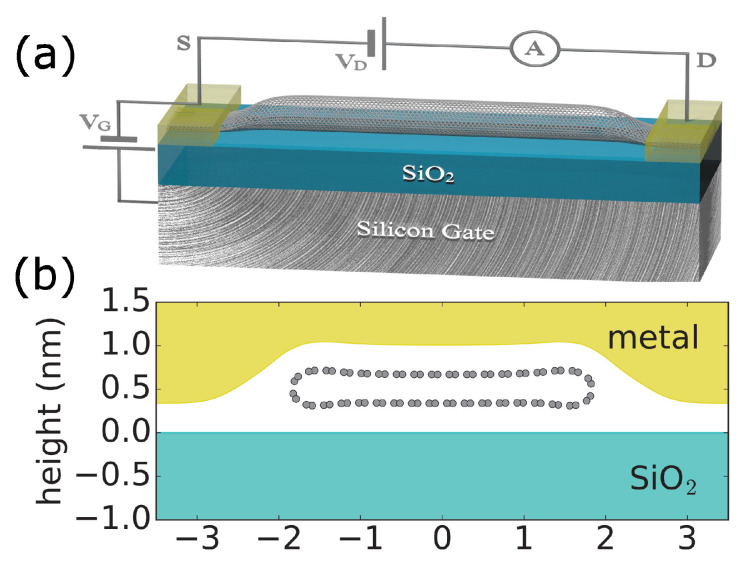
(**a**) Schematics of the single CNT field effect transistor. A thin layer of silicon oxide separates the nanotube from the conducting silicon gate. The deposited metals establish a side electrical contact with the nanotube; such electrons can travel a certain transfer length distance in CNT under the metal before transferring to CNT. (**b**) The calculated geometry of the relaxed (18, 18) CNT under the metal. The CNT collapses due to an interplay of metal surface energy and mutual interactions between metal, substrate, and CNT.

**Figure 2 nanomaterials-13-01774-f002:**
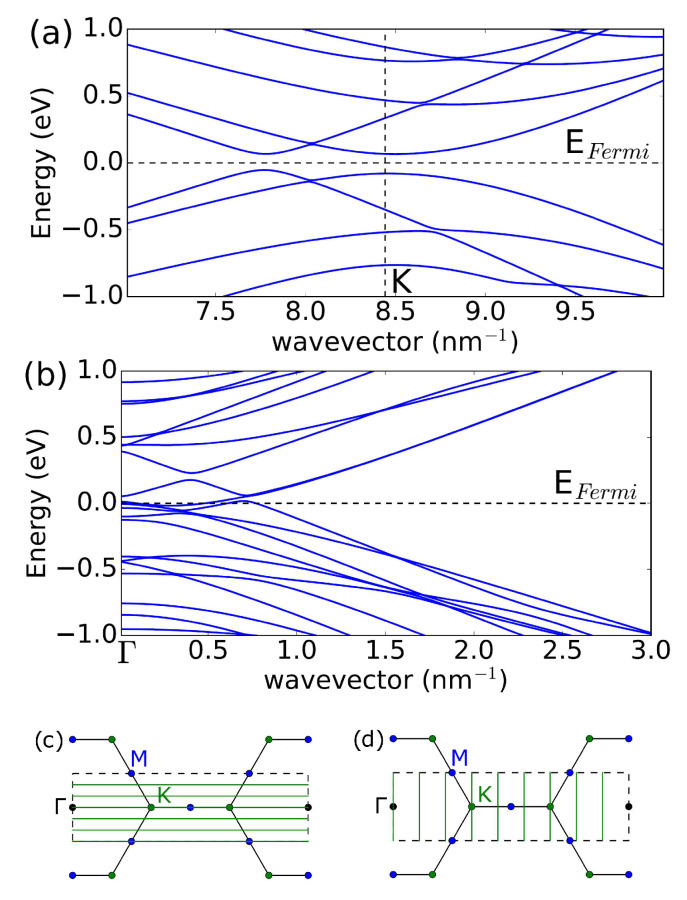
DFT band structures of CNTs distorted by metal: (**a**)—(18, 18), (**b**)—(30, 0). The vertical dashed line in (**a**) shows the position of the *K*-point in the ideal CNT. The horizontal dashed lines in (**a**,**b**) show the position of the Fermi level in charge-neutral CNTs at room temperature. In (18, 18) CNT, the 120 meV bandgap opens, while in (30, 0) CNT, deformations lift the double degeneracy of the bands and result in band crossing. (**c**,**d**) show the 2D graphene Brillouin zone confined by the black dashed rectangle of the area $8\pi^{2}/\left(\sqrt{3}a^{2}\right)$, where a=2.46 Å is the graphene lattice constant. Black, blue, and green circles show high-symmetry k-points Γ, *M*, and *K*, respectively. In the case of armchair CNT, the horizontal 1D green lines in (**c**) correspond to armchair CNT k-points, such that the conduction band minimum and valence band maximum occur at finite 1D wavevector, as shown in (**a**). In the case of zigzag CNT, the vertical 1D green lines in (**d**) correspond to zigzag CNT k-points, such that the conduction band minimum and valence band maximum occur at the zero 1D wavevector, as shown in (**b**). The spacing between the green lines in both (**c**,**d**) equals Δk=2/d, where *d* is the CNT diameter.

**Figure 3 nanomaterials-13-01774-f003:**
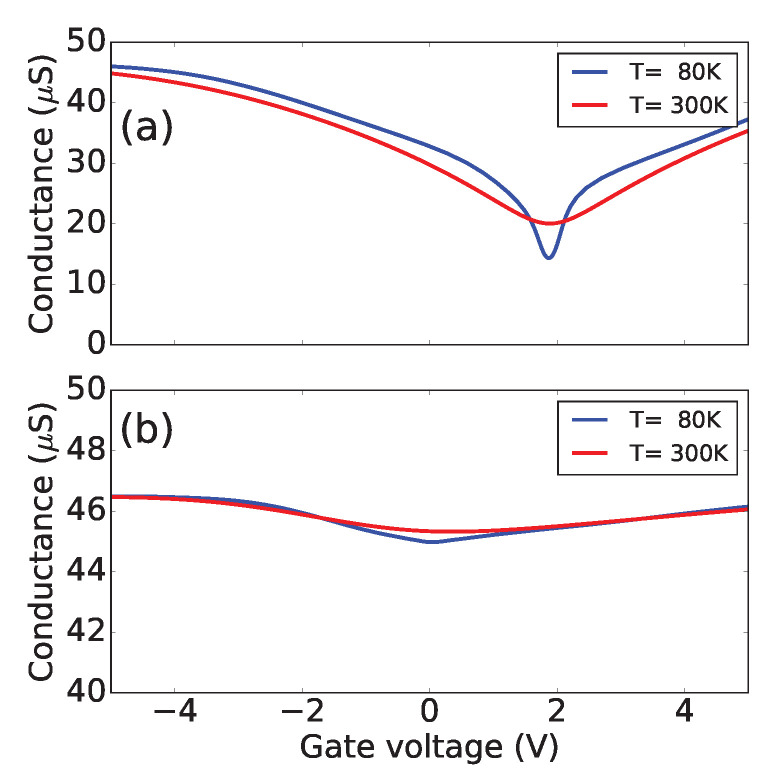
Calculated quasi-ballistic conductance of a CNT-based FET as a function of the gate voltage. (**a**) Under the metal, (18, 18) CNT deformation leads to the bandgap opening shown in Figure 2a, resulting in strong conductance modulation. In the case of (30, 0) shown in (**b**), there is no predicted bandgap opening, and as a result, only a few % conductance modulation is calculated in metallic CNTFETs. The calculations are done at low (80 K) and room (300 K) temperatures in both cases to show that the conductance minimum does not follow Arrhenius temperature-activated behavior.

## Data Availability

The data that support the findings of this study are available from the corresponding authors upon reasonable request.

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
