# Peer review of "Metal Contact Induced Unconventional Field Effect in Metallic Carbon Nanotubes"

_nanomaterials, 2023, doi:10.3390/nano13111774_

Round 1

Reviewer 1 Report

This paper presents a significant contribution to the understanding of the electrical contacts behavior in CNTs. The comprehensive model demonstrates that the structural deformations of CNTs under metals can lead to different current-voltage characteristics. The authors' predictions regarding the conductance of armchair CNTFET can be significantly modulated as a result of the bandgap opening under metal. The deformation of zigzag metallic CNTs results in a band crossing without a corresponding bandgap opening. I think this paper is considerable for publication after addressed the following major issues.

1.      The authors should provide the full meaning of each abbreviation when it appears in the text for the first time, for example, DFT.

2.      The difference between armchair CNTFETs and zigzag metallic CNTs should be further emphasized.

3.      The authors should provide additional information regarding the limitations of their comprehensive model and clearly highlight the differences between their model and other existing models.

4.      The authors should place additional emphasis on the current-voltage characteristics resulting from structural deformations of CNTs.

5.      The figures require modification to enhance their clarity and better convey their intended message.

6.      The authors should carefully review the manuscript and make appropriate edits to improve the clarity, accuracy, and overall quality of the content.

Author Response

We thank all three Referees for making critical comments and questions. Below, we provide point-by-point responses to all comments made by Referee 1.

The referee comments are quoted in full in italic fonts.

Referee’s Comment #1: This paper presents a significant contribution to the understanding of the electrical contacts behavior in CNTs. The comprehensive model demonstrates that the structural deformations of CNTs under metals can lead to different current-voltage characteristics. The authors’ predictions regarding the conductance of armchair CNTFET can be significantly modulated as a result of the bandgap opening under metal. The deformation of zigzag metallic CNTs results in a band crossing without a corresponding bandgap opening. I think this paper is considerable for publication after addressed the following major issues.

Response #1: We thank the Referee for carefully reading our manuscript, making valuable suggestions to improve the presentation and the positive comments on our work.

       Referee’s Comment #2: The authors should provide the full meaning of each abbreviation when it appears in the text for the first time, for example, DFT.

Response #2: We thank the Referee for pointing out the abbreviation issues. It is now fixed.

 Referee’s Comment #3: The difference between armchair CNTFETs and zigzag metallic CNTs should be further emphasized.

Response #3: We thank the Referee for the suggestion. In the manuscript’s revised version, we further emphasize the difference in symmetry of the electronic wavefunctions in zigzag and armchair carbon nanotubes.

Referee’s Comment #4:    The authors should provide additional information regarding the limitations of their comprehensive model and clearly highlight the differences between their model and other existing models.

Response #4: This is an excellent point. We added a discussion of the limitations of our model. The primary limitation is the absence of the electron correlation effects, which are beyond the standard DFT approach and require GW approximation.

Referee’s Comment #5:    The authors should place additional emphasis on the current-voltage characteristics resulting from structural deformations of CNTs.

Response #5: We predict that in the case of large-diameter armchair CNTs, the strong dependence of conductance on the gate voltage should be observed due to the CNT deformation under the metal electrodes. In the conventional picture, the density of states is independent of energy, such that gate voltage-induced Fermi level change should not translate to a change in the conductance. Therefore, a simple measurement of a transfer curve of an armchair CNT field effect transistor can provide an experimental signature of the deformation of the CNT under metallic contacts.

Referee’s Comment #6:  The figures require modification to enhance their clarity and better convey their intended message.

Response #6: We thank the Referee for the suggestion. We modified Fig. 2 and 3, as well as Fig. 1, 2, and 3 captions, to add more clarity to the paper’s main results.  

Referee’s Comment #7: The authors should carefully review the manuscript and make appropriate edits to improve the clarity, accuracy, and overall quality of the content.

Response #7:

We followed the Referee’s suggestion, proofread the manuscript, and fixed typos.

Reviewer 2 Report

The authors compute the effect of deformation of CNTs under metallic contacts on the conductance of field-effect transistors. Numerical DFT simulations are performed for both large-diameter armchair and zigzag nanotubes. The final results can be understood based on the bandstructure modification of CNTs under metal contacts. However, the results are presented in an incomplete manner so that their relevance for readers is limited. More specifically:

- the bandstructures in fig. 2 should be displayed over the first Brillouin zone. In fig. 2 one does not know along which direction in the reciprocal space is the bandstructure represented. In addition, to emphasize the magnitude of the deformation effect of the CNT on the bandstructure, the initial (for nondeformed CNT) and final (deformed) bandstructures should be represented.

- some words describing the working/principles at the basis of the software used are necessary. 

- a variation of the density of states/local potential along the CNT would be welcome

- in p. 6 it is stated that the "electron and hole density of states...are found from the DFT calculation, shown in Fig. 2" No such density of states appear in fig. 2

- a comment on the influence of the type of metal/substrate on the results is needed. The results valid for Pd/SiO2 would be much influenced by other choices of metal/substrate ?

- a comment on the asymmetry of conductances in Fig. 3 with respect to the polarity of the gate voltage is needed

Author Response

We thank all three Referees for making critical comments and questions. Below, we provide point-by-point responses to all comments made by Referee 2.

The Referee comments are quoted in full in italic fonts.

Referee’s Comment #1: The authors compute the effect of deformation of CNTs under metallic contacts on the conductance of field-effect transistors. Numerical DFT simulations are performed for both large-diameter armchair and zigzag nanotubes. The final results can be understood based on the bandstructure modification of CNTs under metal contacts.

Response #1: We thank the Referee for carefully reading our manuscript and making valuable suggestions to improve the presentation of our work.

Referee’s Comment #2: However, the results are presented in an incomplete manner so that their relevance for readers is limited. More specifically:

- the bandstructures in fig. 2 should be displayed over the first Brillouin zone. In fig. 2 one does not know along which direction in the reciprocal space is the bandstructure represented. In addition, to emphasize the magnitude of the deformation effect of the CNT on the bandstructure, the initial (for nondeformed CNT) and final (deformed) bandstructures should be represented.

Response #2: We thank the Referee for the suggestion to include more details of the Brillouin zone and band structure of the undeformed CNT in Fig. 2 to introduce the subject to a wide Nanomaterials audience.

Referee’s Comment #3: - some words describing the working/principles at the basis of the software used are necessary. 

Response #3: We thank the Referee for the suggestion to include some introduction to the Density Functional Theory method and its limitations.

Referee’s Comment #4: - a variation of the density of states/local potential along the CNT would be welcome

Response #4: This is indeed a very good point to discuss. However, our simulations use either infinitely long deformed or undeformed CNTs. We are adding a discussion of the length scales for potential variation near the metal-CNT contact.

Referee’s Comment #5: - in p. 6 it is stated that the “electron and hole density of states...are found from the DFT calculation, shown in Fig. 2” No such density of states appear in fig. 2

Response #5: We thank the Referee for pointing out our sloppiness. In the revised version of the manuscript, we explain how the density of states is obtained for our electrostatics simulations using the bandstructure shown in Fig. 2.

Referee’s Comment #6: - a comment on the influence of the type of metal/substrate on the results is needed. The results valid for Pd/SiO2 would be much influenced by other choices of metal/substrate ?

Response #6: We thank the Referee for the suggestion. In the revised manuscript, we are adding the discussion of the expected results using a different choice of metals. In our simulations, a choice of metal provides the input for the work function difference between the metal and the CNT and the wetting properties. If a metal with stronger adhesion to the substrate is used, the deformations and the effect on the electronic band structure will be stronger. The choice of the substrate can also play a role in that respect. The simulations are done using parameters for the most common metal Pd used for metal contacts and SiO2 substrate.

Referee’s Comment #7: - a comment on the asymmetry of conductances in Fig. 3 with respect to the polarity of the gate voltage is needed

Response #7: We thank the Referee for the comment. In the revised manuscript, we are adding a discussion of the asymmetry due to the asymmetry in the density of states and the finite work function difference between the metal and the CNT.

Reviewer 3 Report

nanomaterials-2339235

 Review

The MS entitled “Metal Contact Induced Unconventional Field Effect in Metallic Carbon Nanotubes” by G. Fedorov and coworkers reports on a study based on Carbon Nanotubes using the DFT (Density Functional Theory). In the paper, deformed CNTs under metal contacts are simulated to study the gate voltage dependence of the conductance of metallic armchair CNTs with deformations. Authors study CNTs with two different chiralities: armchair and zigzag.

In the armchair case it was found that “the transfer characteristics show an ON/OFF ratio of about a factor of two nearly independent of temperature” (lines 75 and 76, page 3), while this behavior was not found in zigzag CNTs.

1 – Please, could authors explain how transfer characteristics were obtained? To obtain the transfer characteristics a charge carrier transport model needs to be considered and a source-to-drain bias voltage applied in order to obtain drain current vs. gate-to-source voltage (transfer characteristics). Equations (1)-(2) are not enough to obtain the transfer characteristics of a FET, but the . May be authors write “transfer characteristics” when they really refer to magnitudes like G(Vg)? Please, clarify this as in transistors “transfer characteristics” unambiguously refer to drain current vs. gate-to-source voltage.

I have some additional questions/remarks:

2 – In the work, Pd was used as the metal in the study. How results would change if a different metal was used? Results would just shift with the value of metal workfunction? For instance, how the value of the bandgap opening at the K-point in the armchair would be modified when the metal is replaced by a different one? Please, could authors comment on that in the paper?

3 – In line 56 (page 2) the units of d, nm?, are missing.

4 – Finally, if authors cannot provide experimental results, could at least use the software employed in the present work to simulate a real CNT device in order to validate simulations?

Author Response

We thank all three Referees for making critical comments and questions. Below, we provide point-by-point responses to all comments made by Referee 3.

The Referee comments are quoted in full in italic fonts.

Referee’s Comment #1: The MS entitled “Metal Contact Induced Unconventional Field Effect in Metallic Carbon Nanotubes” by G. Fedorov and coworkers reports on a study based on Carbon Nanotubes using the DFT (Density Functional Theory). In the paper, deformed CNTs under metal contacts are simulated to study the gate voltage dependence of the conductance of metallic armchair CNTs with deformations. Authors study CNTs with two different chiralities: armchair and zigzag.

In the armchair case it was found that “the transfer characteristics show an ON/OFF ratio of about a factor of two nearly independent of temperature” (lines 75 and 76, page 3), while this behavior was not found in zigzag CNTs.

Response #1: We thank the Referee for carefully reading our manuscript and making valuable suggestions to improve the presentation of our work.

Referee’s Comment #2:

1 – Please, could authors explain how transfer characteristics were obtained? To obtain the transfer characteristics a charge carrier transport model needs to be considered and a source-to-drain bias voltage applied in order to obtain drain current vs. gate-to-source voltage (transfer characteristics). Equations (1)-(2) are not enough to obtain the transfer characteristics of a FET, but the . May be authors write “transfer characteristics” when they really refer to magnitudes like G(Vg)? Please, clarify this as in transistors “transfer characteristics” unambiguously refer to drain current vs. gate-to-source voltage.

Response #2: We thank the Referee for the comment. We agree with the Referee that we don’t perform self-consistent CNTFET simulation for the geometry shown in Fig. 1. Instead, we are solving self-consistently electrostatics in CNT under the metal and calculating the number of channels available for transport to find G(Vg). This approach is valid in a quasi-ballistic regime, which we assume in our simulations. In the revised version of the manuscript, we explain our approach better and clarify that we predict that measured transfer characteristics on an actual CNTFET should look similar to the reported curves in Fig. 3.

Referee’s Comment #3:

I have some additional questions/remarks:

2 – In the work, Pd was used as the metal in the study. How results would change if a different metal was used? Results would just shift with the value of metal workfunction? For instance, how the value of the bandgap opening at the K-point in the armchair would be modified when the metal is replaced by a different one? Please, could authors comment on that in the paper?

Response #3: We thank the Referee for the suggestion. In the revised manuscript, we are adding the discussion of the expected results using a different choice of metals. In our simulations, a choice of metal provides the input for the work function difference between the metal and the CNT and the wetting properties. If a metal with stronger adhesion to the substrate is used, the deformations and the effect on the electronic band structure will be stronger. The choice of the substrate can also play a role in that respect. The simulations are done using parameters for the most common metal Pd used for metal contacts and SiO2 substrate.

Referee’s Comment #4:

3 – In line 56 (page 2) the units of d, nm?, are missing.

Response #4: We thank the Referee. Yes, d is nm. It is fixed now.

Referee’s Comment #5:

4 – Finally, if authors cannot provide experimental results, could at least use the software employed in the present work to simulate a real CNT device in order to validate simulations?

Response #5: This comment is related to Comment #2 above. The main message of our paper is that we predict conductance dependence with the gate voltage in nominally metallic CNTFETs. The Referee is correct that the transfer characteristics are expected to differ slightly if the full simulations are performed, but the paper’s central message will not be affected. In the revised version of the manuscript, we are discussing the limitation of our approach. For example, in the full CNTFET ballistic simulations, transmission coefficient T(E) would have energy dependence, which is not included in our model. However, the bandstructure effect would lead to a much stronger G(Vg) dependence than T(E) variation.

Round 2

Reviewer 1 Report

NA

Author Response

We thank the Referee for the positive review of our manuscript. 

Reviewer 2 Report

The authors have responded in an appropriately manner to the suggestions made in the first review report. The paper can now be published

Author Response

(The authors gave the same response as above.)
